# Global Warming and Long-Distance Spread of Invasive *Discoglossus pictus* (Amphibia, Alytidae): Conservation Implications for Protected Amphibians in the Iberian Peninsula

**DOI:** 10.3390/ani12233236

**Published:** 2022-11-22

**Authors:** Dani Villero, Albert Montori, Gustavo A. Llorente, Núria Roura-Pascual, Philippe Geniez, Lluís Brotons

**Affiliations:** 1Forest Sciences and Technology Centre of Catalonia (CTFC), Carretera de Sant Llorenç¸ de Morunys km 2, 25280 Solsona, Spain; 2Center for Ecological Research and Forestry Applications (CREAF), 08193 Cerdanyola del Vallès, Spain; 3Centre de Recerca i Educació Ambiental de Calafell (CREAC), GRENP/Aj, 43882 Calafell, Spain; 4Departament de Biologia Evolutiva, Ecologia i Ciències Ambientals, Facultat de Biologia, Universitat de Barcelona, Av, Diagonal, 643, 08028 Barcelona, Spain; 5Departament de Ciències Ambientals, Facultat de Ciències, Universitat de Girona, 17007 Girona, Spain; 6CEFE, Univ Montpellier, CNRS, EPHE-PSL University, IRD, Biogéographie et Ecologie des Vertébrés, 34293 Montpellier, France; 7Spanish National Research Council (CSIC), 08193 Cerdanyola del Vallès, Spain

**Keywords:** Amphibia, Discoglossidae, invasive species, climatic change, niche overlap, conservation

## Abstract

**Simple Summary:**

*Discoglossus pictus* is a North African amphibian that was introduced in southern France early in the 20th century and has spread south and north along the Mediterranean coastal plains up to 170 km. Many studies have demonstrated that *D. pictus* competes against native species with similar breeding strategies, pointing out abiotic conditions as the main driver tipping the balance in favor of one or another species. This study aims to assess the impact of the spread of *D. pictus* on native *Iberian Discoglossus* and other native species, analyzing the potential roles of long-distance dispersal and long-term climate warming in the Iberian Peninsula. The study area covers the western Mediterranean region, including all *Discoglossus* species in northwestern Africa, Sicily, the Iberian Peninsula, and southern France. Our results show a strong climatic niche overlap between *D. pictus* and targeted species in the Iberian Peninsula, including endemic *Discoglossus* species. Future projections of climatic change suggest that climatic suitability will increase for all species, both inside and outside the Natura 2000 network, with the only exception being a moderate and widespread decrease for *Pelodytes punctatus.* However, these positive trends are reversed within Natura 2000 sites where most species are explicitly targeted, jeopardizing the effectiveness of protected areas in a long-distance dispersal scenario.

**Abstract:**

*Discoglossus pictus* is a North African amphibian that was introduced in southern France early the 20th century and has spread south and north along the Mediterranean coastal plains up to 170 km. In order to disentangle the conservation implications of the spread of *D. pictus* for sensitive native species, we examined the impact of long-term climate warming on the basis of niche overlap analysis, taking into account abiotic factors. The study area covered the distribution ranges of all genus *Discoglossus* species in northwestern Africa (659,784 km^2^), Sicily (27,711 km^2^), the Iberian Peninsula, and southern France (699,546 km^2^). Niche overlap was measured from species environmental spaces extracted via PCA, including climate and relief environmental variables. Current and future climatic suitability for each species was assessed in an ensemble-forecasting framework of species distribution models, built using contemporary species data and climate predictors and projected to 2070′s climatic conditions. Our results show a strong climatic niche overlap between *D. pictus* and native and endemic species in the Iberian Peninsula. In this context, all species will experience an increase in climatic suitability over the next decades, with the only exception being *Pelodytes punctatus*, which could be negatively affected by synergies between global warming and cohabitation with *D. pictus.*

## 1. Introduction

The introduction of invasive species is, after habitat destruction, the second most important cause of biodiversity loss on Earth [1,2,3]. The main problems related to the introduction of invasive species are competition with local fauna, introduction of pathogens, and genetic pollution of autochthonous populations [2,4,5,6]. Guijarro et al. estimated that in the last 300 years, 39% of all known extinctions have been driven by invasive species [7].

*Discoglossus pictus* is a North African species that was introduced from Algeria to Europe in Banyuls Sur Mer (southern France) in the early 20th century [8]. Currently, the invaded area of *D. pictus* extends over a continuous range from southern France to the northeastern Iberian Peninsula, occupying more than 7000 km^2^ [9,10] (Figure 1). Evidence of the spread of *D. pictus* in southern Europe shows a similar rate of expansion as other invasive amphibians [10]. Montori et al. and Llorente et al. demonstrated that populations of *D. pictus* have moved 60 km west and 140 km south within the Iberian Peninsula, with good prospects for progression through climatically favorable regions [11,12,13,14,15]. Furthermore, unforeseen establishment of *D. pictus* in the metropolitan area of the city of Barcelona (Spain) [13] has highlighted the proneness of the species for human-mediated long-distance “stowaway” dispersal [16], thus boosting the risk of accelerating range expansion. Ongoing climatic changes at the global scale can presumably play a critical role in this acceleration [17], especially since future climate conditions for amphibians in the Iberian Peninsula are predicted to approach current conditions found in North Africa [18].

Species interactions between invasive *D. pictus* and native species have been consistently assessed [15,19,20,21,22,23,24,25]. Specifically, *D. pictus* shows a strong biotic niche overlap with *Pelodytes punctatus* and *Epidalea calamita*; three species with similar phenology and breeding strategies. Moreover, *D. pictus* has been identified as an asymptomatic carrier of *Batrachochytrium dendrobatidis*, increasing the exposure of other sympatric amphibians to chytrid zoospores [26]. Competition between native and non-native species can drive recipient communities to become less structured [20,24], and abiotic conditions have been identified as the main driver exacerbating this process [19]. Beyond the invaded area, potential interaction with the Iberian-endemic *Discoglossus galganoi* highlights the importance of predicting the limits of the potential expansion of *D. pictus*, particularly in a long-distance dispersal scenario based on its ability for long-distance “stowaway” dispersal. The progression of *D. pictus* in the Iberian Peninsula can have potential impacts on the populations of the *D. galganoi* eastern subspecies *D. galganoi jeanneae*, which is listed as “vulnerable” in the Spanish Red List [27].

In order to assess the impact of the spread of *D. pictus* on native Iberian *Discoglossus* species and other sensitive species, in this paper we examined the potential roles of long-distance dispersal and long-term climate warming in the Iberian Peninsula. Our aims were: (1) to assess niche overlap between *D. pictus* and a group of target species from the Iberian Peninsula with similar biotic requirements, including Iberian congeneric species *D. galganoi galganoi* and *D. galganoi jeanneae*, as well as Iberian native species *E. calamita*, *P. punctatus*, *Pelodytes ibericus* (Iberian endemism with the same niche requirements as *P. punctatus* [27] and, thus, potentially sensitive to a long-distance dispersal of *D. pictus*), and *Discoglossus scovazzi* (included to assess niche overlap with a parapatric species with large shared boundaries in eastern Morocco); (2) to predict changes in climatic suitability due to global warming for Iberian native species and *D. pictus* in the Iberian Peninsula; (3) to assess potential impacts of a long-distance dispersal scenario on Iberian native species under different protection regimes throughout the full distribution ranges of native Iberian species. Protection regimes were based on Natura 2000, which is the official network of protected areas of the European Union and covers the most valuable and threatened species and habitats. Insights from these investigations are needed to formulate long-term conservation strategies.

## 2. Materials and Methods

### 2.1. Study Area

The study area covered the western Mediterranean region, including distribution ranges of *D. pictus* and *D. scovazzi* in northwestern Africa (659,784 km^2^), *D. pictus* in Sicily (27,711 km^2^), and the Iberian Peninsula and southern France (699,546 km^2^). Corsica and Sardinia were excluded from the analysis because they are both occupied by endemic *Discoglossus* species (*D. montanelli* and *D. sardus*, respectively) with no contact zones with the *D. pictus* range.

Protection regimes within the Iberian Peninsula and southern France were defined on the basis of the Natura 2000 GIS database [28]. In this area, Natura 2000 includes 781 sites covering nearly 20% of the territory. Among these sites, 189 are explicitly targeted to protect Habitats Directive Annex II species *D. galganoi galganoi* (167 sites) and *D. galganoi jeanneae* (22 sites). Supplementary conservation targets are also defined in Natura 2000 sites with 69 sites extending protection to *E. calamita* (Habitats Directive Annex IV and Bern Convention), 66 sites to *P. punctatus* (Bern Convention), and 3 sites to *P. ibericus* (Iberian endemic species).

### 2.2. Species Data

Native-range data for *D. pictus* included 76 occurrences from Algeria and Tunisia (presence-only data: [29] and unpublished data from *P. geniez*) and 115 occurrences from Sicily (among 373 10 × 10 km atlas cells: [30]). Northern African data also provided 113 occurrences for *D. scovazzi* from Morocco (presence-only data: [31] and unpublished data from *P. geniez*) (Figure 1).

European invaded-range data for *D. pictus* included 59 occurrences from southern France (presence-only data: [9,32] and 59 occurrences from Spain (among 7720 10 × 10 km atlas cells: [33]) (Figure 1). Spanish data also provided occurrences for *D. galganoi jeanneae* (500), *D. galganoi galganoi* (860), *E. calamita* (2463), *P. punctatus* (790), and *P. ibericus* (261). Finally, Portuguese data completed the Iberian distribution for *D. galganoi galganoi* (414)*, E. calamita* (583), and *Pelodytes* spp. (242) (among 1166 10 × 10 km atlas cells: [33]). In the herpetological atlas of Portugal, *Pelodytes* spp. included different *Pelodytes* species (*P. punctatus*, *P. ibericus*, and *P. atlanticus*), all of which had overlapping distribution ranges with no clear boundaries [34]. Due to the lack of objective criteria to split the data among the different *Pelodytes* species, Portuguese occurrences were assigned to the more widespread species, *P. punctatus*, based on the assumption of a strong abiotic niche overlap between all *Pelodytes* species in the Iberian Peninsula [34,35].

### 2.3. Environmental Data

Following Sillero, five explanatory variables connected to the abiotic ecological requirements of amphibians were selected for subsequent analyses [36]. Temperature extremes, annual precipitation, and relief are relevant features for amphibian distribution in the Mediterranean region. These factors were summarized in 2 relief predictors, altitude and slope, and in 3 climate predictors: maximum temperature of warmest month (BIO5), minimum temperature of coldest month (BIO6), and annual precipitation (BIO12). Climate and altitude layers were downloaded from the WorldClim 1.4 repository ([37], www.worldclim.org accessed on 15 August 2022) at 5 min spatial resolution (~9 × 9 km), and slope was derived from WorldClim DEM at 30 s spatial resolution.

For long-term climatic suitability analyses, we also downloaded climate layers for the late 21st century (averaged climate projections for the 2061–2080 time period) based on the Fifth Assessment IPCC report [38] for both the most optimistic (RCP 2.6) and pessimistic (RCP 8.5) greenhouse gas scenarios. Following recent studies dealing with the impacts of climate change on biodiversity [39,40], we averaged six different Global Climate Models (CNRM-CM5, IPSL-CM5A-LR, HadGEM2-ES, MPI-ESM-LR, GISS-E2-R, and CCSM4) to reduce uncertainties, producing one single layer for each climate predictor and greenhouse gas scenario.

### 2.4. Niche Overlap Analysis

To search for similarities/dissimilarities in environmental conditions between ranges of *D. pictus* and target species from the Iberian Peninsula, we conducted a PCA analysis [38]. The first two principal components of the PCA were used to visualize the variation patterns of native species and invaded species ranges in a bivariate plot. Environmental spaces were delimited using maximum convex polygons, including all species occurrences, and the intersection of spaces between species was used as a measure of niche overlap. The analyses were conducted in R, using the packages ade4 and gpclib [41,42].

### 2.5. Climatic Suitability Changes

Climatic suitability was modeled for all species by running five widely used niche-based modeling algorithms implemented in the biomod2 platform [43]. These models included: (1) generalized linear model (GLM), (2) generalized additive model (GAM), (3) multivariate adaptive regression splines (MARS), (4) generalized boosting model (GBM), and (5) flexible discriminant analysis (FDA). All models were trained on the basis of species presence-absence samples and current climatic conditions, and then the models were projected to future climatic scenarios. The predictive performance of each model was assessed by means of the relative operating characteristic (ROC) curve and area under the curve (AUC) [44], on the basis of a subsampling approach that randomly split a 70% subset of the sample for model building and the remaining 30% for testing predictions. Models were also replicated 10 times using environmental stratified presence-absence subsamples built from available species occurrences to determine a more robust estimate of the predictive performance from the averaged AUC of the replicated cross-validations [45]. For presence-only data from northern Africa and southern France, this meant using the known surveyed areas in each region [46] as the environmental background to stratify subsampling, thus obtaining pseudo-absence samples of equal size as the presence-only samples [47]. On the other hand, original presence-absence data from Italy, Spain, and Portugal were subsampled to balance environmental gradients within the training samples [47]. Environmental stratification was based in the WWF Terrestrial Ecoregions of the World [48]. AUC measurement is independent of the threshold at which the model’s prediction is considered and values range from 0 to 1; AUC scores close to 1 mean perfect model predictions, while AUC scores close to 0.5 indicate predictions no better or worse than random. The potential problems raised by Lobo et al. on the use of AUC as a measure of model performance were considered to be minor because AUC was used to rank models obtained from the same dataset and within the same geographical area according to their predictive performance [49]. We applied an ensemble-forecasting framework by computing a consensus of single-model projections (from models with AUC > 0.7 using AUC values as model weights) using a weighted average approach [50].

To determine the potential range of affectation of *D. pictus* in the Iberian Peninsula, native-range predictions were built using North African and Sicilian data and projected to new geographical areas in southern Europe. These models were built using only occurrence data from the native range where the species was in equilibrium with the environment. Here, agreement between observed and predicted distribution within the invaded range was assessed using AUC scores, which were computed using invaded-range occurrence samples [51].

Differences between current and future climatic suitability scores were used to measure changes in climatic suitability, yielding positive changes when future scores were larger than current scores, and negative changes in the opposite case. To allow comparisons between species, scale values from current and future climatic suitability predictions were homogenized using standard scores (i.e., difference between raw climatic suitability score and mean climatic suitability score, divided by the climatic suitability standard deviation). This normalization implied adjusting raw scale values (0–1) to a common scale that quantified the number of standard deviations above (or below) the mean. Hence, species-specific misalignments between current and future climatic suitability raw scales were corrected to achieve accurate measurements of climatic suitability changes based on standard scores. Since future forecasts of climatic suitability were based on current climatic suitability models, these misalignments were consistently corrected by using the mean and standard deviation from current climatic suitability predictions to compute future climatic suitability standard scores. Calculations of climatic suitability changes for sensitive species were circumscribed to species occurrence actual data. On the other hand, *D. pictus* computations were not only restricted to occurrence data in the actual invaded range, but assumed a conjectural successful long-distance dispersal scenario based on demonstrated *D. pictus* proclivity for human-mediated long-distance dispersal [16], which will drive widespread distribution of the species throughout the Iberian Peninsula by the end of 21st century, with overlapping ranges with all Iberian targeted species. This means that we defined the future potential range of *D. pictus* as all occurrence localities where any of the Iberian native species were recorded. Invaded and potential ranges of *D. pictus* allowed us to examine differences in climatic suitability across targeted species and under different protection regimes by distinguishing areas outside and inside Natura 2000, as well as Natura 2000 sites where targeted species were explicitly protected.

## 3. Results

### 3.1. Niche Overlap Analysis

The main two axes of the PCA accounted for 81% of the total variance: PC1 (45% of variance) was negatively correlated with minimum temperature and precipitation (r = −0.58 and r = −0.53, respectively) and positively correlated with altitude (r = 0.52); PC2 (36%) was positively correlated with maximum temperature (r = 0. 51) and negatively correlated with precipitation and slope (r = −0.47 and r = −0.43, respectively). Comparisons of the environmental ranges of *Discoglossus* congeneric species indicated that the environmental conditions in their native ranges were quite similar (Figure 2), suggesting that congeneric species could easily occupy broader extensions without the existence of biotic interactions. A similar situation was found when comparing the native and invaded ranges of *D. pictus*, with an invaded range that seemed to be a smaller subset of the conditions occupied in the native range. After plotting the occurrences of *D. pictus* in a bivariate plot of the first two factors, only four occurrences (3.3% of occurrences) from the invaded range were outside the native environmental space, showing little niche shifting to new environmental conditions from the invaded area (Figure 2). Non-congeneric native species also showed a strong niche overlap with *D. pictus*, with more than 80% of each species’ environmental space intersecting with the *D. pictus* environmental space within the invaded range.

### 3.2. Long-Term Climatic Suitability Changes

Climate ensemble native-range predictions efficiently captured the climate envelope of *D. pictus* (mean AUC = 0.92, st. dev. AUC = 0.01), even when they were compared to the invaded range in southern Europe (mean AUC = 0.83) (Figure 3). Climate ensemble outcomes from other species also exhibited excellent predictive performance, with AUC > 0.9 for all species except the most widespread *E. calamita* (mean AUC = 0.79, st. dev. AUC = 0.01) and *D. galganoi galganoi* (mean AUC = 0.80, st. dev. AUC = 0.02).

We found contrasting amphibian responses to climate change according to the examined greenhouse gas scenarios. Climatic suitability changes from the pessimistic scenario showed potential negative responses for all species, stressing that against extreme climate warming, *D. pictus* invasion might be the less important problem facing native species (Figure 4). In contrast, the optimistic scenario yielded positive widespread responses to climate warming for all species. Only *P. punctatus* showed a general loss of climatic suitability, including areas inside and outside the Natura 2000 network within the current invaded range of *D. pictus*, but also in the potential range based on the long-distance dispersal scenario (Figure 4). No major differences were reported for species among *D. pictus* dispersal scenarios and protection regimes based on paired t-tests, except for Natura 2000 sites where *D. galganoi galganoi* (*p* < 0.05) and *E. calamita* (*p* < 0.05) were explicitly located within the potential range of *D. pictus* (Figure 4).

## 4. Discussion

The native-range predictions showed that *D. pictus* spread in southern Europe is following the best environmental paths. Large areas of the southern Iberian Peninsula are highly suitable for the establishment of *D. pictus*, and our results suggest that this will be exacerbated in the long term (Figure 3). Several studies have tested the value of niche modeling for assessing the risk of amphibians from another region invading a given area [52,53], highlighting the benefits of using native-range data to assess the geographical potential of invasive species in the face of climate change [54,55,56]. The potential species distribution is projected based on the assumption that current niches from native distributions reflect species’ environmental preferences, which are retained in the invaded new areas [57,58]. However, it is important to note that the native-range models were calibrated using a limited set of occurrence data and probably underestimate the potential distribution of *D. pictus* in the Iberian Peninsula, thereby representing conservative predictions of the real potential distribution of the species.

All indications suggest that *D. pictus* is not at equilibrium with environmental conditions in Europe, with an invasion front expanding at equal rates and similar strategies as other invasive amphibians. In fact, long-distance “stowaway” dispersal has been reported for other amphibians, such as *Rhinella marina* in Australia, emphasizing that the magnitude of dispersal through anthropogenic transport should not be underestimated [16]. Importantly, the proneness to human-mediated dispersion means that the species is likely to reach most parts of potential invaded ranges at frequencies high enough to set up new populations if conditions are suitable. The establishment, persistence, and expansion of *D. pictus* in Barcelona province [12,13] provides strong evidence supporting this hypothesis. Current climatic suitability predictions show that such areas may exist in the southern Iberian Peninsula, albeit separated from the main part of the current *D. pictus* invaded range by 200 km of land with low climatically suitability, but with high densities of transportation networks. Hence, high rates of anthropogenically assisted transport may overcome limited dispersal abilities of ectotherms [59], suggesting that *D. pictus* may be ultimately able to colonize any part of the Iberian Peninsula that provides conditions suitable for population persistence [53].

To determine the pattern of spread of *D. pictus* in the Iberian Peninsula in the late 21st century, we predicted climatic suitability under optimistic and pessimistic climate warming scenarios. At best, our results indicate that the whole Mediterranean basin in the Iberian Peninsula may experience an increase in *D. pictus* climatic suitability under optimistic future climatic conditions (Figure 4). On the other hand, the pessimistic scenario showed very poor conditions for the spread of *D. pictus*, but also for the persistence of all native species included in the analysis and, hence, the results were unhelpful for invasive *D. pictus* risk assessment (Figure 4). It is widely thought that climate change will exacerbate problems with invasive species [60], but the many ways in which changes could affect the ranges of species and the complex interactions that could potentially facilitate or hinder shifts make accurate predictions very difficult [61]. Additionally, the ways in which climatic variables will interact in the future may be quite different from the current situation. Such changes are certain to influence environmental suitability and invasiveness in ways other than simply through altered climatic tolerance. Thus, projections of potential future distributions need to be interpreted with caution. Our approach is based on the assumption that climate is the major driving factor of species distribution [62], and that analysis of the climatic preferences of species can therefore be used to predict areas where species could occur at regional scales. Although climate sets the broad limit of ectothermic species range, other factors such as hydrology, disturbance regime, competition, and other biotic interactions determine the presence or absence of a species in a particular area and at finer local scales [15,19,20,21,22,23,24,25,63]. The question is whether such simplifications enable useful projections under climate change. A number of studies have empirically demonstrated that carefully implemented bioclimatic models can recover the broad-scale direction of species range changes under climate change [64,65,66]. Range changes measured from niche models have important intrinsic uncertainty, as changes are contingent on several unmeasured factors. However, evidence shows that models can recover the tendency towards range increases or decreases with reasonable accuracy. Thus, one possible approach to limit uncertainty is to conservatively interpret model projections. By quantifying relative climatic suitability changes for each species, we avoid making quantitative inferences about population parameters, such as changes in range, abundance, or extinction risk, that are not explicitly modeled [67]. Other limitations of this study include uncertainties inherent in climate-change scenarios and coarse resolution of anomalies in GCM [68]. The coarse resolution of the data used to build the models (10 × 10 km) may also mask some fine-scale variations in species’ ecological requirements that were not detectable at the spatial scale of our analysis. Because the influence of each environmental variable in determining the species’ niche is scale dependent, different degrees of ecological niche variation can arise among populations depending on the spatial resolution of the analyses [69].

Based on our results, the combined effects of global warming and the spread of *D. pictus* should not pose a risk for most species in the recipient amphibian communities in the next decades, even if, hypothetically, endemic congeneric species face new long-distance dispersal events by invasive *D. pictus*. Specifically, we show that in an optimistic scenario, climate conditions at the end 21st century are likely to become more suitable for *D. pictus* and native species inside and outside the protected areas. Only non-endangered *P. punctatus* will likely be negatively affected by synergies between global warming and cohabitating with *D. pictus*. However, these general trends are reversed for native species within targeted Natura 2000 sites. Hence, explicitly designated protected areas would preserve species worse than unprotected areas, jeopardizing the effectiveness of protected areas in a long-distance dispersal scenario. If true, *D. pictus* long-distance dispersal events stress the importance of implementing early detection and monitoring plans within Natura 2000 sites having high-suitability for the species. The approach we present here can provide insights into the basic mechanisms underlying range expansion and inform efforts to focus preventive monitoring in the areas that are more at risk [70], but also evaluate and adapt Natura 2000 site-specific targets [71] in order to effectively preserve endangered species.

The ecological impacts of invasive amphibians primarily involve direct changes to single native communities, populations, or species [15,20,22,23,24,25], but community-level impacts also logically result from the loss of, or dramatic declines in, native populations [5]. The strong niche overlap between *D. pictus* and native species suggests that climatic suitability changes might involve decreases in the abundance of native species [72], driving a worst case scenario for local extinctions and geographical range contractions [73,74]. However, most feasible impacts encompass the simplification and homogenization of amphibian communities due to *D. pictus* establishment [15,20,22,23,24,25]. Regarding impacts on congeneric species, competitive exclusion between *D. pictus* and *D. scovazzi* in northern Africa and between *D. galganoi galganoi* and *D. galganoi jeanneae* in the Iberian Peninsula reveals that niche overlap goes beyond abiotic factors and drawing clear borders allows parapatric coexistence between congeneric neighbors [75,76,77,78]. Our results do not provide any evidence against this finding.

Amphibians are among the species of highest conservation concern due to their widespread decline worldwide [79,80], but their poor representation in conservation strategies may only be understood by the resilience of many species in human-dominated landscapes [81,82,83,84]. Nevertheless, many studies have demonstrated that the impacts of global changes are able to boost the fragility of the group due to the combined effects of global warming, invasive species, and emerging diseases [85,86,87]. In our case, *D. pictus* is an asymptomatic carrier of *B. dendrobatidis* and sympatric *E. calamita* has been shown to shed increased amounts of *B. dendrobatidis* zoospores [26]. These impacts should not be neglected, and appropriate monitoring and conservation planning based on strong national and international (e.g., EU) nature conservation policies may help to ensure species conservation and the integrity of native communities.

## Figures and Tables

**Figure 1 animals-12-03236-f001:**
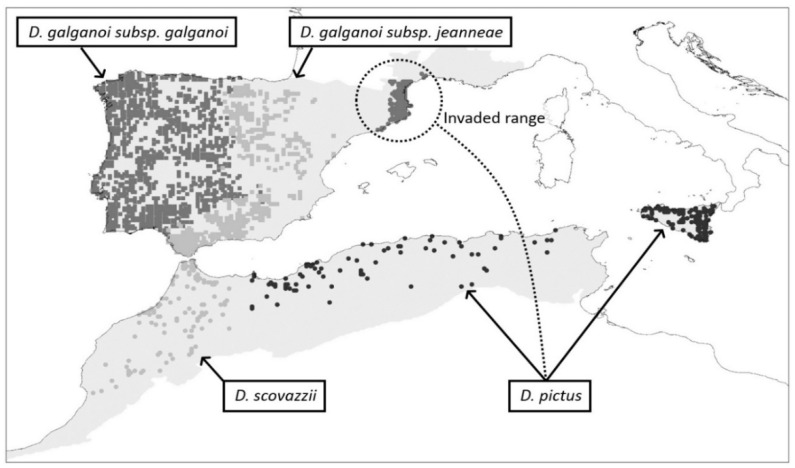
Distribution of *Discoglossus pictus* and congeneric and parapatric species within the study area (shaded) in the western Mediterranean region.

**Figure 2 animals-12-03236-f002:**
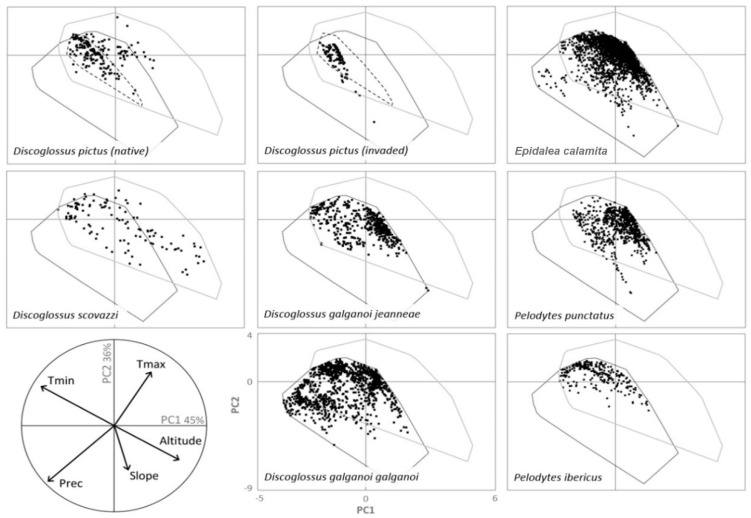
Environmental space of *Discoglossus* native and invaded (only *D. pictus*) species ranges in a bivariate plot of two principal components. The convex hulls show the global climatic space in North Africa (solid grey), Iberian Peninsula, southern France (solid black), and Sicily (dashed black, only for *D. pictus*). The correlation circle indicates the importance of each bioclimatic variable to the two principal axes of the Principal Components Analysis (PCA), which jointly explain 81% of the variance in the data.

**Figure 3 animals-12-03236-f003:**
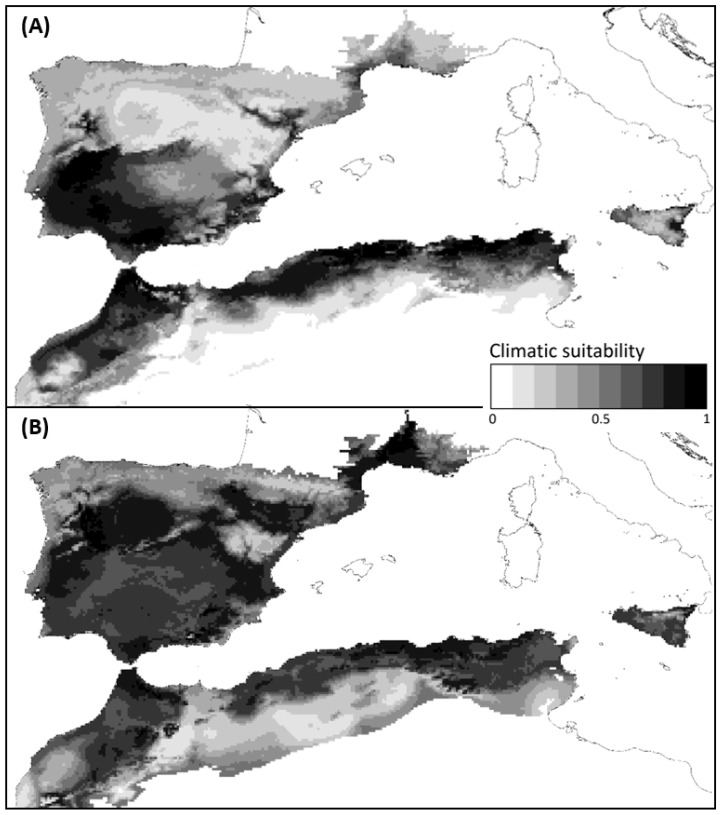
Current (**A**) and future (**B**) climatic suitability predictions for *Discoglossus pictus* in northern Africa and southern Europe. Future climatic predictions are based on the optimistic greenhouse gas scenario (RCP 2.6).

**Figure 4 animals-12-03236-f004:**
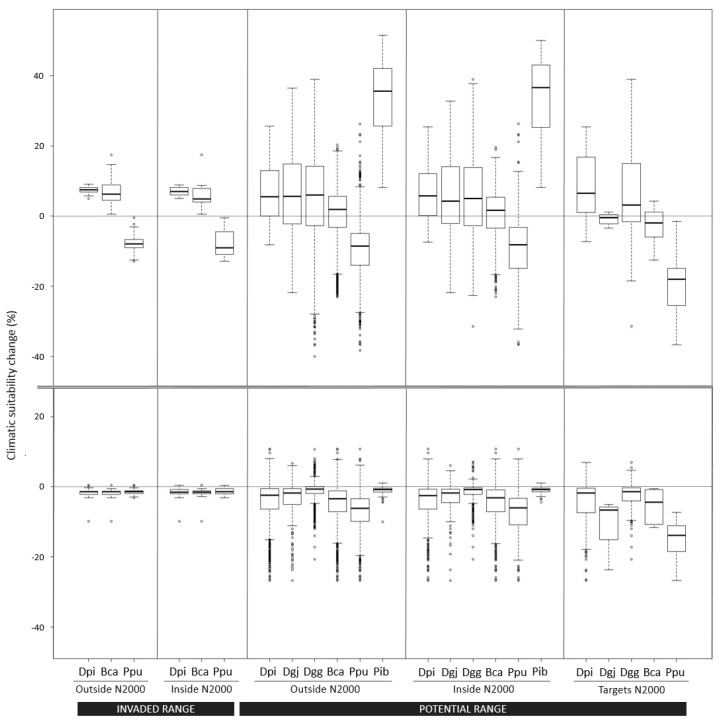
Species climatic suitability change (%) under optimistic (**upper**) and pessimistic (**bottom**) climate warming scenarios, different *D. pictus* ranges (invaded or potential), and protection regimes, splitting areas inside and outside Natura 2000 (N2000) from those Natura 2000 sites where species are explicitly targeted (Targets N2000). Dpi, *Discoglossus pictus*; Dgj, *Discoglossus galganoi jeanneae*; Dgg, *Discoglossus galganoi galganoi*; Bca, *Epidalea calamita*; Ppu, *Pelodytes punctatus*; Pib, *Pelodytes ibericus*. Points show outliers; lower and upper whiskers indicate the 5% and 95% percentiles, respectively; lower and upper hinges indicate the 25% and 75% quartiles, respectively; and the central black line indicates the median value.

## Data Availability

Not applicable.

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
