# Peer review of "Global Warming and Long-Distance Spread of Invasive Discoglossus pictus (Amphibia, Alytidae): Conservation Implications for Protected Amphibians in the Iberian Peninsula"

_animals, 2022, doi:10.3390/ani12233236_

Round 1

Reviewer 1 Report

The study justification and choice of taxa needs to be carefully explained. The text needs to be thoroughly reviewed; I've made linguistic suggestions and pointed out inconsistencies that need to be fixed.

Reviewer 2 Report

Authors,

Your manuscript provides a deep insight into the combined effect of future climate changes and invasive species on the conservation of native Alytidae frogs. Your manuscript is very detailed, and I commend that. The only thing I found that needs to be addressed is some typos and some stylistic aspects.

In summary, I think this manuscript will make a fine contribution to Animals and suggest to be accepted.

Reviewer 3 Report

The work of Villero and colleagues try to investigates the effect of global warming on the changes in climatic suitability of Discoglossus pictus and Iberian native species of amphibian. Furthermore, they try to assess potential impacts to these species under different protection regimes in a long-distance dispersal scenario and long-term climate warming. They present a niche overlap between D. pictus and congeneric/native species of amphibian, which allows to focus only on the abiotic factor. The data and modelling used for the experimental design allow to show that climatic suitability will increase for the majority of these species inside and outside Natura 2000 network. This approach can help in the development of effective management strategies to preserve endangered species.   The work is nicely done and the manuscript well writed. For my part, I need clarification/improvement on two points :   1) On the 2.3. environmental data part (l.124) there is no information about which year correspond to the climate data downloaded from WorldClim database. The species data where collected in an interval of time between 1996 from now. Does the climate data and species data collect are in the same range of time period?   2) Concerning the niche overlap analysis, the authors use a PCA visualisation to show that « Comparisons of the environmental ranges of Discoglossus congeneric species indicate that the environmental conditions in their native ranges are quite similar ». To confirm this similarity an additional statistic analysis can be applied, like a between–classes inertia analysis (Doledec et al., 2000) with a Monte-Carlo randomizations.

Reviewer 4 Report

This is an interesting study, well designed and executed, and I think it should be published. However, I have some concerns that I think must be addressed before publication.

1) The manuscript needs extensive review of English language and style. I have pointed some instances needing improvement on my line by line comments below. The rest of the document should be thoroughly reviewed for clarity and concision.

2) There are some transparency issues in the methods and results. The authors should provide a better description of their distribution modelling, specially regarding their methodology for pseudoabsences. Also, some critical data mentioned is not presented in tables, such as the data in Figure 4. Having these data in tables would greatly facilitate a critical examination of the results by the reviewers. More details on this point can be found on the point by point comments below.

3) My main concern is that the results present do not support the main conclusion. If I understood correctly, the authors interpreted the differences in relative climatic suitability as a indication that the invasive species will not pose a risk to native species. This ignores many other factors that would influence species interactions, such as body size and life history characteristics, which might give invasive species a competitive advantage despite being less adapted to local climates. More important, the authots mention the invasive species is a known asymptomatic vector of Batrachochytrium dendrobatidis, so their presence might cause significant damages to local species, despite being less adapted to local climate. I suggest that the authors either soften these claims or provide a stronger reasoning as to why these conclusions are supported by their results.

Line 19 - "early in the 20th century"
Line 20 - "spreading south"
Line 21 - D. pictus must be in italics
Line 21 - "competes"
Line 23 - "species"
Line 29 - "these positive trends"
Line 52 - "It has been estimated that in the last 300 years, 39% of all known extinctions have been driven by invasive species [7]."
Line 76 - "Although neither native species has serious conservation problems"
Line 146 - "The first two principal components of the PCA"
Line 162 to 165 - Any reference for this methodology?
Line 166 - How many pseudoabsence points for each species? How were they distributed (e.g. minimum and maximum distances allowed from presence points)?
Line 168 - If I understand this correctly, you ran some models with pseudo absences and some with true absences. It might be argued that these models are not comparable, but this can easily be verified by running the true absence models with pseudo absences instead and comparing the results. Including this comparison in your supplemental material would prevent such critiques.
Line 184 - What was the probability of occurrence threshold used to consider a point as a presence in this comparison?
Line 210 - Explain what is the Natura 2000 network
Figure 2 - It is hard to see which is the dotted line, the dots are too small.
Line 232 - Only use "significant" to refer to an actual statistical test, with an associated p-value
Line 233 - It would be great to have a table showing the actual overlap between each pair of species
Line 238 - Provide table with AUCs, Sensitivity and Specificity of each models
Line 244 - Provide table with average and standard deviation of the changes in climatic suitability for each species
Line 253 - Provide table with the results of the t-tests
Figure 3 - Why show the results for 2.6 only? It would be interesting to compare with 8.5, so we can know how different the invasion suitability of regions would be under each scenario
Line 322 - "from niche models"
Line 336 to 352 - This whole paragraph seems unsupported by the results. Nowhere the presence of invasive species was included as a predictor for the distribution of native species. This seems to rely purely on the differences between relative climate suitabilities, which cannot tell us the effect an invasive species could have on the native species. In fact, on the next paragraph the authors tell us that these effects have been recorded and can go beyond what is predicted by abiotic overlaps. I suggested rewriting this section and being more cautious on the claims made here.
Line 371 - This is a very important information and should be mentioned in the introduction.
Line 375 - remove question mark
Line 377 - I don't think the results support this claim.

Line 19 - "early in the 20th century"
Line 20 - "spreading south"
Line 21 - D. pictus must be in italics
Line 21 - "competes"
Line 23 - "species"
Line 29 - "these positive trends"
Line 52 - "It has been estimated that in the last 300 years, 39% of all known extinctions have been driven by invasive species [7]."
Line 76 - "Although neither native species has serious conservation problems"
Line 146 - "The first two principal components of the PCA"
Line 162 to 165 - Any reference for this methodology?
Line 166 - How many pseudoabsence points for each species? How were they distributed (e.g. minimum and maximum distances allowed from presence points)?
Line 168 - If I understand this correctly, you ran some models with pseudo absences and some with true absences. It might be argued that these models are not comparable, but this can easily be verified by running the true absence models with pseudo absences instead and comparing the results. Including this comparison in your supplemental material would prevent such critiques.
Line 184 - What was the probability of occurrence threshold used to consider a point as a presence in this comparison?
Line 210 - Explain what is the Natura 2000 network
Figure 2 - It is hard to see which is the dotted line, the dots are too small.
Line 232 - Only use "significant" to refer to an actual statistical test, with an associated p-value
Line 233 - It would be great to have a table showing the actual overlap between each pair of species
Line 238 - Provide table with AUCs, Sensitivity and Specificity of each models
Line 244 - Provide table with average and standard deviation of the changes in climatic suitability for each species
Line 253 - Provide table with the results of the t-tests
Figure 3 - Why show the results for 2.6 only? It would be interesting to compare with 8.5, so we can know how different the invasion suitability of regions would be under each scenario
Line 322 - "from niche models"
Line 336 to 352 - This whole paragraph seems unsupported by the results. Nowhere the presence of invasive species was included as a predictor for the distribution of native species. This seems to rely purely on the differences between relative climate suitabilities, which cannot tell us the effect an invasive species could have on the native species. In fact, on the next paragraph the authors tell us that these effects have been recorded and can go beyond what is predicted by abiotic overlaps. I suggested rewriting this section and being more cautious on the claims made here.
Line 371 - This is a very important information and should be mentioned in the introduction.
Line 375 - remove question mark
Line 377 - I don't think the results support this claim.
